# Cancer Research Funding in Africa

Oluwasegun Afolaranmi[1], Elise M. Garton[2], Olaoluwa Ezekiel Dada[3], Sehar Salim Virani[4,5], Abdul R. Shour[6], Adedayo A. Onitilo[6] & Syed Nabeel Zafar [5] ✉

## Abstract

**Background** Africa is projected to witness the steepest rise in cancer incidence and mortality in the coming decades. Therefore, it is critical to understand the current landscape of cancer research funding to identify key gaps and inform decision-making.

**Methods** We conducted a retrospective study of funded cancer research projects involving at least one African country over the 20 years between January 2004 and December 2023. Data was collected from four publicly available databases, namely the International Cancer Research Partnership (ICRP), National Institutes of Health World Research Portfolio Online Reporting Tools (WoRLD RePORT), ClinicalTrials.gov (CTG), and International Clinical Trials Registry Platform (ICTRP). We retrieved data on country, year of funding, cancer types, study types, and funding sources. Furthermore, we used incidence, mortality, and prevalence data to compare the level of funded projects to the burden of disease.

**Results** A total of 3047 unique funded projects/grants were reported from all 4 databases, with a consistent rise in the number of funded projects throughout the study period. Egypt and South Africa had the most funded cancer research projects, and 9 (16%) countries had no reported studies. Breast, lung, and cervical cancers received the highest funding allocation. We found that several cancers, notably cervical, prostate, and liver, are relatively underfunded compared to their disease burden. 70% of projects reported in ICRP/WoRLD RePORT were funded by the U.S. NIH. Notably, 40% of studies in CTG/ICTRP reported local funding, with Egypt accounting for 94% of these locally financed studies.

**Conclusions** This study provides a comprehensive overview of the current state of cancer research funding in Africa, highlighting notable gaps and critical insights to guide data-driven decision-making.

## Plain Language Summary

The burden of cancer in Africa is rising sharply; therefore, it is crucial to study the distribution of cancer research funding on the continent to identify important gaps and guide decision-making. In this study, we reviewed 20 years of data from international research databases and identified which African countries and cancer types received funding and how this compared to the actual burden of disease. We found that although there is an increasing trend in cancer research funding in Africa, the funding landscape is unevenly distributed, as very few countries account for most of the funded projects. Moreover, cancers like cervical, prostate, and liver were underfunded relative to their impact. Most funding came from external sources, except in Egypt, with significant local support. These findings highlight the need for more equitable, needs-based cancer research investment in Africa.

Cancer presents a tremendous global health burden, with 20 million new cases and nearly 10 million deaths worldwide in 2022[1]. The disease burden and mortality vary considerably across organ systems, geographical regions, environments, and resource settings. For example, although the incidence rates for breast cancer in North American and European countries can be up to four times higher than in Western African countries, West Africa accounts for some of the highest mortality rates seen with the disease[1]. Importantly, Africa is faced with a rapidly increasing burden of cancer, with the continent projected to experience the sharpest rise in cancer incidence and mortality over the coming decades, compared to other parts of the world[2]. This disproportionate cancer burden in Africa is multifactorial, with inadequate screening services, diagnostic delays, barriers to accessing treatment, and insufficient locally-driven research playing major roles[3–5].

The rise in cancer burden in Africa calls for an unprecedented level of action involving multiple governmental and non-governmental stakeholders within the continent and beyond. To comprehend changing patterns and guide policy aimed at delivering effective, efficient, and equitable care across the entire spectrum of oncology, locally relevant and context-appropriate cancer research is crucial. This, in turn, requires significant funding commitments. Globally, up to 24.5 billion US dollars (USD) was invested in cancer research between 2016 and 2020[6], and investment in cancer research is thought to be rising. However, significant disparities remain in funding distribution globally[7]. Given differences in health care systems and resources, funding for cancer research needs to be more equitable. This is necessary to ensure that scarce resources are directed towards solving the most locally relevant challenges and in the most effective

[1]Cancer Research UK Cambridge Institute, University of Cambridge, Cambridge, UK. [2]Center for Global Health, National Cancer Institute, National Institutes of Health, Rockville, MD, USA. [3]College of Medicine, University of Ibadan, Ibadan, Nigeria. [4]Department of Surgery, Aga Khan University, Karachi, Pakistan. [5]Department of Surgery, University of Wisconsin–Madison, Madison, WI, USA. [6]Marshfield Clinic Research Institute, Marshfield Clinic Health System, 1000 N Oak Ave, Marshfield, WI, 54449, USA. ✉e-mail: zafars@surgery.wisc.edu

manner. Resource allocation and funding decisions rely on the global cancer burden data (GLOBOCAN), which depend on well-established cancer registries. Consequently, the lack of established cancer registries in several African countries, as well as the deficiencies in quality and coverage of the existing registries, means that the data estimates might not adequately reflect the cancer burden on the continent[8]. Therefore, it is imperative to consistently and meticulously track cancer funding data, in order to monitor progress and address deficiencies. Previous works have provided useful insights into the state of cancer research funding in Africa; however, these have either been through content analysis of funding acknowledgements in published studies or have focused on a singular database[6,7,9,10]. Therefore, here we reviewed data from four public databases with both unique and overlapping characteristics to achieve a more comprehensive view of the landscape of cancer research funding on the continent.

Despite the rise in global cancer research investments, only a small fraction of these investments directly addresses cancer challenges in Africa. As highlighted in this study, there is a striking reliance on external funding sources. Additionally, 16% of African countries had no reported funded cancer studies, reflecting critical regional disparities. These findings emphasize the urgent need to align research investments with high-burden cancers such as cervical, prostate, and liver cancers, which, despite their significant impact, remain underfunded relative to their disease burden. Our study unveils critical gaps and provides important data to guide agenda-setting for cancer research funding in Africa.

## Methods
### Study design and data sources
We conducted a retrospective observational study of funded cancer research projects involving at least one African country over the 20 years between January 2004 and December 2023. We reviewed cancer research projects across four large publicly available online databases; International Cancer Research Partnership (ICRP)[11], National Institutes of Health World Research Portfolio Online Reporting Tools (NIH World RePORT)[12], ClinicalTrials.gov (CTG)[13], and the World Health Organization International Clinical Trials Registry Platform (WHO ICTRP)[14]. The databases used for this study were selected due to their well-annotated records of projects, funding-related information, and broad catchment of included projects[15].

The ICRP is a network of 173 international funding organizations whose mission is to enhance global collaboration. Members submit information about their funded projects, such as cancer sites, collaborating partners, and study types, to the ICRP database, which is estimated to contain about 65% of funded cancer research projects globally[11]. Similarly, the NIH World RePORT is hosted by the U.S. National Institutes of Health (NIH) and provides funding data from directly or indirectly funded studies from 17 large biomedical funding organizations[12]. ClinicalTrials.gov is an online library maintained by the U.S. National Institutes of Health (NIH) that holds data on clinical research studies reported by study sponsors and investigators. It includes data on both clinical trials and observational studies from over 200 countries since its launch in the year 2000[13]. Finally, the WHO ICTRP reports clinical trial studies data from 17 primary and 5 partner registries from across the world, including the Pan African Clinical Trials Registry (PACTR)[14].

### Inclusion and exclusion criteria
We included cancer-related research projects conducted between January 2004 and December 2023, and including at least one African country among the study locations/sites. Eligible studies spanned clinical trials, observational studies, epidemiological assessments (e.g., GLOBOCAN data), and funding-related analyses. Projects were excluded if they were non-peer-reviewed (e.g., editorials, commentaries, opinion pieces) or if the articles were not available or not primarily written in English. Research focused on non-cancer topics, including public health or infectious diseases unrelated to oncology, was excluded. Professional development funding from other organizations, like private foundations and think tanks (e.g., conference

attendance, education, and travel grants), was also not included. Duplicate entries were identified through matching project titles, unique project IDs, or NCT numbers and removed during de-duplication. Additionally, studies missing key metadata (e.g., no cancer type or funding source) were excluded from relevant sub-analyses.

### Search strategy
We searched each of the four databases for cancer-related studies conducted in at least one African country, with any form of funding during the study period. Studies could include primary investigators and/or collaborating investigators with an institutional affiliation in any country in Africa. The search strategy included the same keywords across all databases: (cancer*) OR (neoplas*) OR (malignan*) OR (tumor*) OR (tumor*) OR (carcinoma*) OR (oncology*). However, specific adaptations in other search parameters were made to fit the different databases. For the ICRP, all African countries were included together to extract data on funded projects for the continent. Furthermore, the ICRP uses a Common Scientific Outline (CSO) coding system that enables uniform classification of study types into "Biology", "Etiology", "Prevention", "Early Detection, Diagnosis and Prognosis", "Treatment", and "Cancer Control, Survivorship and Outcomes Research". Therefore, we also extracted data on project counts and the relative proportion of projects (reported as percentage relevance) for each CSO category in ICRP. Similarly, for NIH World RePORT, we used "Africa" as a filter to extract data for all African countries at once. For CTG and WHO ICTRP, each African country was inputted individually, with date filters from 01 January 2004 to 31 December 2023. The NIH World RePORT database only offers data from 2016 onwards; therefore, our date filters were from January 2016 to December 2023 for this database. As estimates of the burden of disease, we obtained data on age-standardized incidence and mortality rates as well as 5-year prevalence for specific cancer types in Africa, and for individual African countries, from the International Agency for Research on Cancer (IARC) Global Cancer Observatory (GLOBOCAN) 2022 data[16]. Search results were exported as ".*csv*" files in most cases, or manually curated into an Excel file where necessary.

### Data de-duplication, aggregation, and analysis
Data on the number of funded studies, years of funding, study types, cancer types, and funding sources were extracted for each country across the four databases. To improve the accuracy of our data and avoid over-reporting, we performed several de-duplication steps. Both ICRP and World RePORT included individual records for each funded year of a project. For ICRP data, base projects were de-duplicated from total projects by unique project ID. Similarly, for World RePORT, the downloaded data were de-duplicated by unique project title and only individual projects in their first year of funding were used for downstream analysis.

While there are likely duplicate projects reported across databases from several funders, the NIH-funded projects in both ICRP and World RePORT were sourced from the same underlying database and make up a large proportion of both databases. Therefore, we deduplicated NIH-funded projects across both databases using the unique 8-digit project number. In the same vein, because the ICTRP features investigator-reported clinical projects from multiple registries, including the CTG, we used unique NCT IDs to deduplicate projects across both databases.

Cancer types were largely reported as presented in the databases. However, for uniformity across databases and simplicity of data reporting and analysis, we grouped related cancer types where necessary. Bone cancers and soft tissue sarcomas were grouped as "Bone/Soft tissue", cancers of the oral cavity and lip, nasal cavity and paranasal sinuses, pharynx, larynx, and salivary glands grouped as "Head and Neck", non-thyroid endocrine or neuroendocrine tumors grouped as "Other Endocrine/NET", and non-stomach/colorectal or unspecified gastro-intestinal tumors as "Other GI, Peritoneum". Where mentioned, the term "Viral-associated cancers" was used to refer to cancers with etiology prominently linked to viral agents as defined in the literature[17]. These include cancers of the Cervix, Head and Neck, Liver, Anus, and

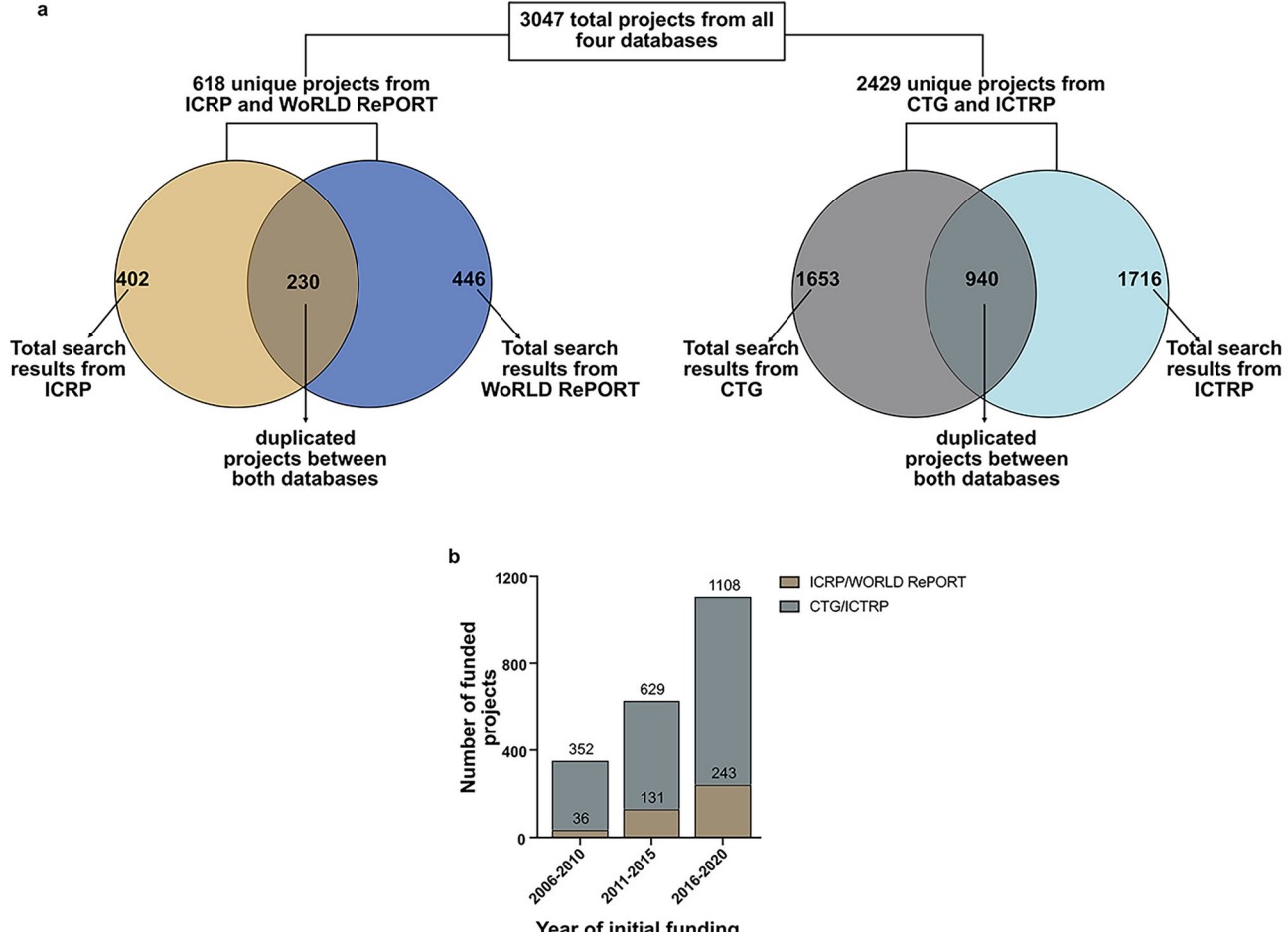

**Fig. 1 | Total number of studies and 5-year trends of funded cancer projects in African countries across all four databases. a** Total search results from each database through the 20-year study period are shown. Deduplication of overlapping projects between ICRP/World RePORT and between CTG/WHO ICTRP resulted in 3047 projects across all databases taken forward for subsequent analyses. **b** A 5-year trend in the number of cancer projects from all four databases between 2006 and 2020. ICRP International Cancer Research Partnership, WoRLD RePORT National Institutes of Health World Research Portfolio Online Reporting Tools, CTG ClinicalTrials.gov, ICTRP World Health northwest rocOrganization International Clinical Trials Registry Platform, N.B. The NIH World RePORT database only includes data from 2016 onwards.

Stomach, as well as Non-Hodgkin's Lymphoma and Kaposi's Sarcoma. Where a project lists more than one cancer type, each cancer type was recorded individually; therefore, the total number of cancer types exceeds the number of projects.

Funding source data were reported as defined by the ICRP and NIH World RePORT databases. For CTG and WHO ICTRP, where the types of funding agencies were not categorized, we manually grouped the funding sources into four categories based on internet verifications of agency/names listed under the funder/sponsor columns. These included "Industry/Pharma", "Investigator/self", "International/foreign organization/institution", and "Local organization/institution". Furthermore, we compared trends in funded cancer research projects to the burden of cancer in Africa. Data on the specific amount of funding received for projects was limited. Of the 4 databases studied, only the NIH World RePORT provides some data on funding amount; however, these data were only available for less than 15% of the African studies reported. Therefore, we have not included an analysis of funding amounts here. Descriptive statistics, graphical representations, and figures were made using Prism GraphPad (v10) and Affinity Designer (v2).

### Reporting summary

Further information on research design is available in the Nature Portfolio Reporting Summary linked to this article.

## Results

### Overview of funded cancer research projects in Africa

We identified 402 funded projects from the ICRP, 446 from WoRLD RePORT, 1653 from CTG, and 1716 from the ICTRP database over the 20 years studied (Fig. 1a). Following deduplication by unique NIH 8-digit IDs, we identified 230 projects overlapping between both ICRP and WoRLD RePORT, resulting in 618 unique projects between these two databases (Fig. 1a, left). Similarly, querying by unique NCT IDs revealed 940 overlapping projects between CTG and ICTRP, resulting in 2429 unique projects from these databases (Fig. 1a, right). Altogether, our search generated 3047 unique funded projects across all 4 databases (Fig. 1a). Notably, there was a consistently rising trend in the number of funded projects throughout the study period, with the total studies reported from all databases nearly doubling every 5 years between 2006 and 2020. The number of studies reported in the ICRP and World RePORT databases rose from 36 in the years 2006–2010 to 243 between 2016 and 2020. Similarly, CTG and ICTRP reported 352 funded studies from 2006 to 2010, compared to 1108 from 2016 to 2020 (Fig. 1b).

To determine the regional distribution of cancer research funding, we grouped funded projects within and across all databases according to the United Nations African sub-regions (Fig. 2). Projects could have researchers in multiple countries and African sub-regions. Nearly half (49.6%) of all projects were from countries in Northern Africa, with Central African

**Fig. 2 | Distribution of funded cancer projects across African subregions. a** Total funded studies (%) from all 4 databases per UN subregion, with **b** showing contributions from the different databases to each region. N.B. The sum of the numbers shown in A here (3611) is more than the total number of studies in Fig. 1a (3047), as some studies involve multiple countries cutting across different subregions.

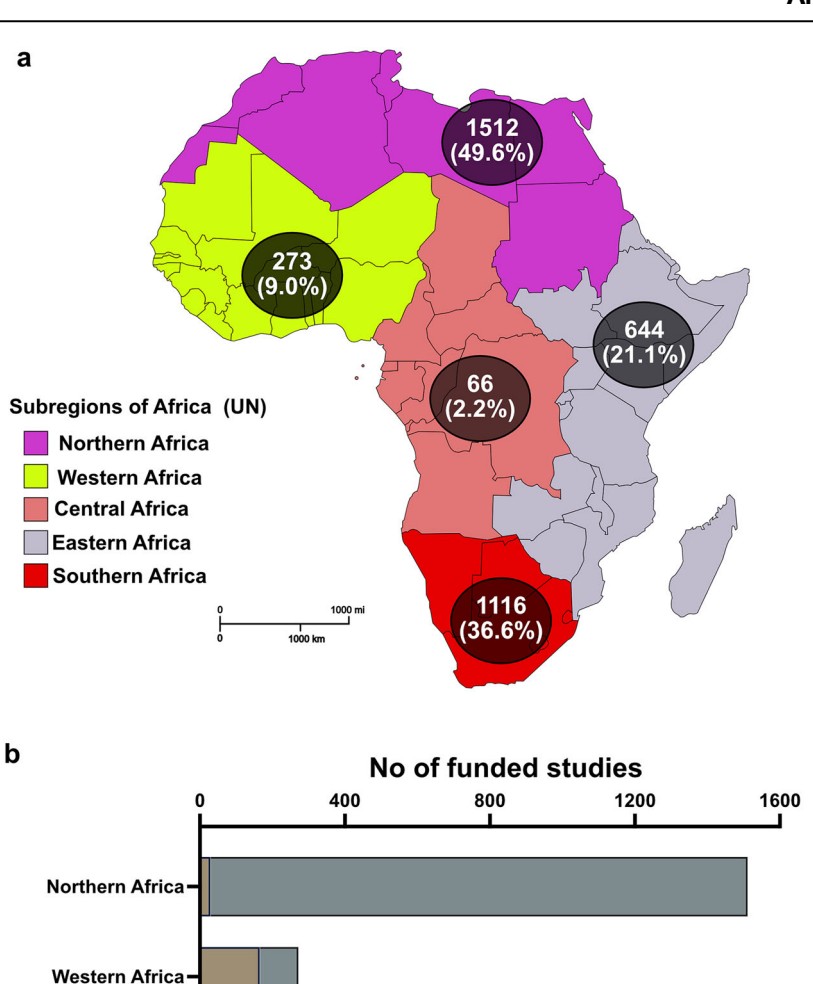

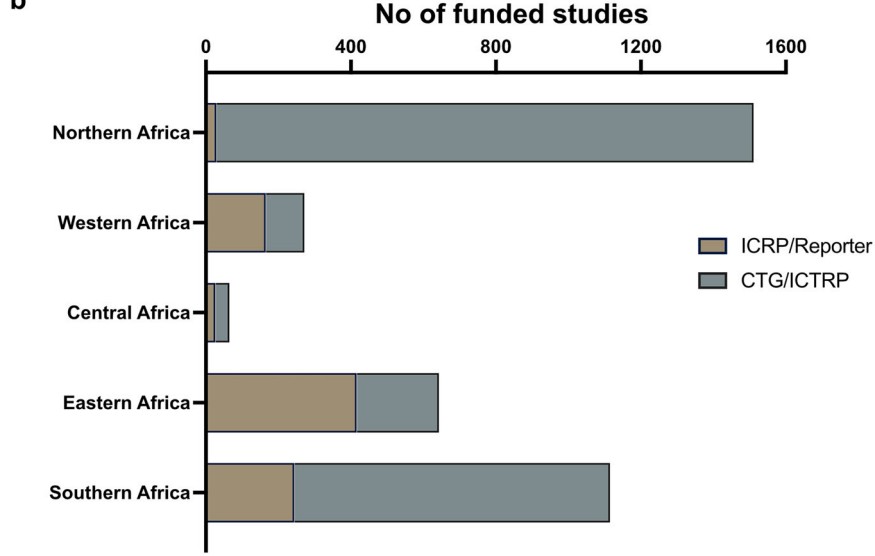

countries accounting for only 66 projects (2.2%) (Fig. 2a). Moreover, we found remarkable differences in regional contributions from different databases, suggesting variations in funding flow, study types, partnerships, and research architecture. For example, most reported studies from Northern (98%) and Southern (81%) Africa were pooled from the CTG and ICTRP databases that prominently feature clinical trials and commercial and/or locally sponsored research. Conversely, ICRP/World RePORT accounted for most studies identified from Eastern (65%) and Western (61%) African regions (Fig. 2b). The total number of funded projects through the 20-year study period ranged from 0 to 1300 for individual African countries, with a mean of 63.35 and a median of 9 projects. At least one of 5 countries (Egypt, South Africa, Kenya, Uganda, and Nigeria) featured in 90% of all reported funded projects, and 9 (16%) of the 57 countries had no reported studies across all four databases (Fig. 3).

### Funded projects by cancer type and study type
We next examined the distribution of cancer types investigated. Across all databases, Breast (627), Lung & Pleural (364), Cervix Uteri (267), Head and

Neck (237), and Liver & Biliary (199) cancers had the highest total numbers of funded projects (Table 1). A notable number of projects (355) did not study a specific cancer type. Furthermore, the relative proportions of investigated cancers were largely similar between different databases (Table 1). Notably, cancers highly associated with viral infections, such as Cervix Uteri, Liver, Non-Hodgkin's Lymphoma, and Kaposi's Sarcoma, featured prominently among the funded projects.

The CTG and WHO ICTRP databases categorize projects into either interventional or observational study types. Across both databases, the majority of the reported projects (80%) were interventional (Fig. 4a). CTG provides additional granular data on the study designs for reported projects. From these, we found that 79% of the interventional studies reported were randomized clinical trials, and all reported observational projects were prospective cohort studies. The ICRP database classifies reported studies into either "Clinical Trial", "Research" or "Training" project types, with some studies including a combination of these. Most of the reported studies from Africa fell into the Research category (302), followed by Clinical Trial (128), with fewer studies on Training (37) (Fig. 4b). The ICRP CSO allows

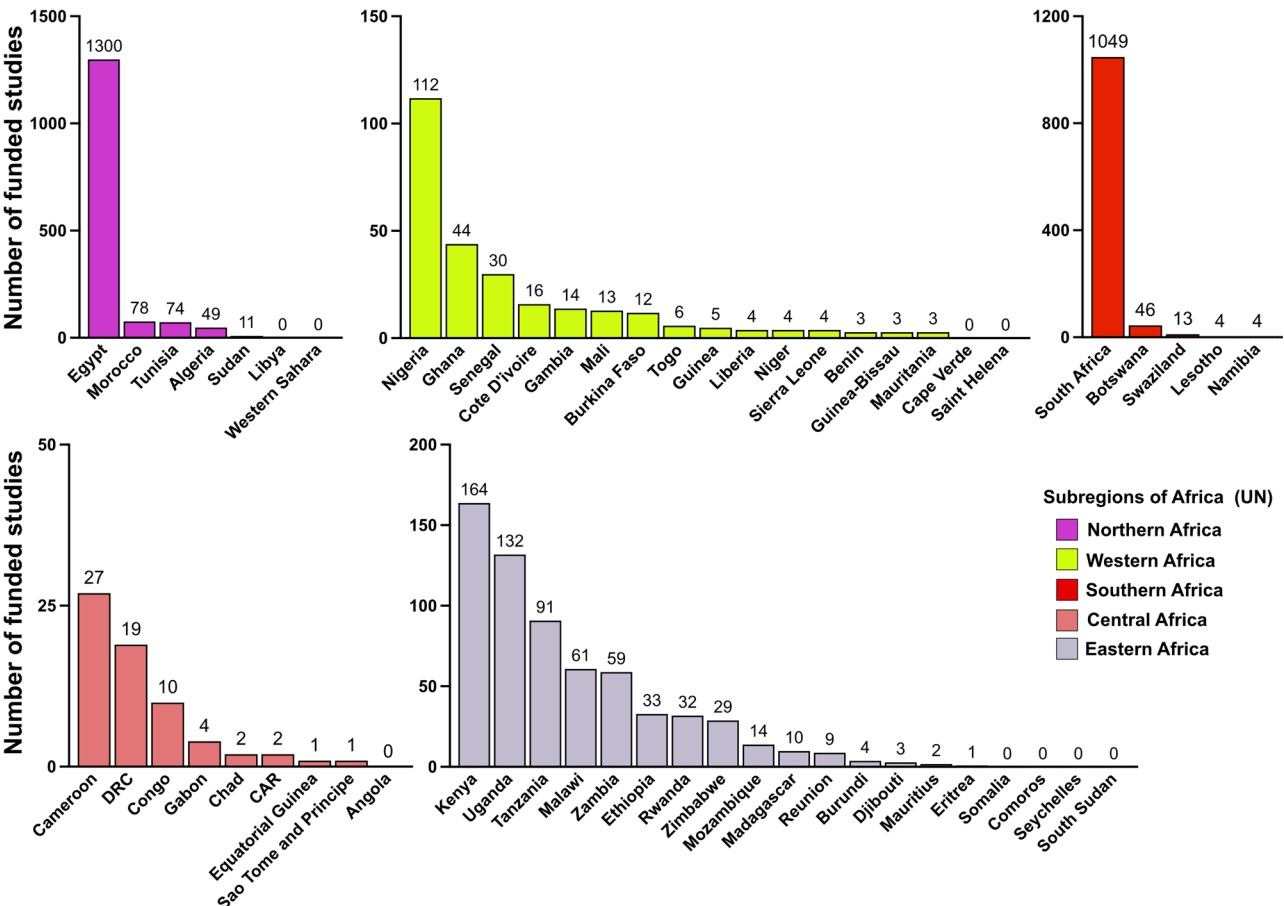

**Fig. 3 | Country-specific distribution of funded cancer studies.** Total number of funded cancer projects for individual African countries, grouped by UN subregions.

further classification of funded projects into six relevant scientific domains. Over the 20-year period in ICRP, 23.1% of funded projects in Africa were on causes of cancer/etiology, 21.5% on early detection, diagnosis, and prognosis, 18.3% on cancer prevention, 13.6% on cancer control, survivorship, and outcomes, 13.1% on treatment, and 10.4% on biology (Fig. 4c, Left).

To identify the magnitude and direction of changes, if any, in the prioritization of these scientific areas over the study period, we trended CSO data over consecutive 5-year periods from 2006 to 2020 (Fig. 4c, Right). The relative proportion of funded studies on the causes/etiology of cancer increased nearly twofold from 12.50% during the 2006–2010 period to 21.83% in the 2011–2015 period and 23.65% in 2016–2020. Studies on cancer treatment also rose from 5.45% (2006–2010) to 13.42% (2016–2020). Conversely, the percentage relevance of studies on cancer control, survivorship, and outcomes dropped by half, from 25.07% (2006–2010) to 12.46% (2016–2020).

### Funding sources for cancer research projects in Africa
The ICRP and World RePORT feature projects driven by international partnerships, therefore the vast majority (84%) of studies in these two databases were funded by foreign governments (Fig. 5a). The US National Institutes of Health (NIH) funded 70% of African cancer research projects reported in ICRP/World RePORT, predominantly via the National Cancer Institute (NCI) and the Fogarty International Center (FIC) (Fig. 5a). Of note, the Cancer Association of South Africa (CANSA), a local organization, funded 12% of studies reported in ICRP/World RePORT (Fig. 5a). On the other hand, CTG and WHO ICTRP databases present investigator-reported clinical studies, therefore, we grouped the funders/sponsors in both databases into four categories (Fig. 5b). Strikingly, 40% of CTG/ICTRP projects were sponsored by local organizations or institutes such as

government agencies, hospitals, universities, and research institutes in African countries. However, a closer look at this data revealed that locally funded projects across both databases were almost exclusively from Egypt (94%) (Fig. 5b). Industry-funded research was also prominent across these two databases, accounting for 43% of funded projects. Ten percent of projects were funded by international or foreign public or philanthropic organizations, and 6% were reported as self-funded.

### Correlation between funded cancer research and disease burden
The relationship between the number of funded studies and the incidence and mortality for specific cancer types is shown in Fig. 6. Breast cancer has the highest incidence and mortality and appropriately represents the largest share of funded research projects. Lung and Head & Neck cancers appear to be positive outliers, with a high number of funded projects relative to their disease burden in Africa. In contrast, cancers such as cervix uteri, prostate, liver and biliary, stomach, bladder, esophagus, and thyroid have high incidence and mortality but are underfunded relative to their disease burden.

### Discussion
This study highlights the increasing trend in cancer research funding in Africa over the past 20 years, identifying notable disparities in the distribution of projects across regions and cancer types. Our findings underscore the importance of more equitable funding distribution and the need for greater investment in local research infrastructure and talent development to reduce reliance on external sources. By combining data across four large databases with unique attributes, this study provides the most comprehensive analysis of cancer research funding trends and patterns across Africa to date.

**Article**

**Table 1 | Cancer types investigated by funded projects**

|  | ICRP | CTG/ICTRP | TOTAL |
|---|---|---|---|
| BREAST | 80 | 547 | 627 |
| LIVER (AND PLEURAL) | 104 | 260 | 364 |
| NOT SITE-SPECIFIC CANCER | 168 | 187 | 355 |
| CERVIX UTERI | 143 | 124 | 267 |
| HEAD & NECK | 124 | 113 | 237 |
| LIVER (AND BILLIARY) | 88 | 111 | 199 |
| COLON & RECTAL | 38 | 138 | 176 |
| NON-HODGKIN'S LYMPHOMA | 96 | 58 | 154 |
| LEUKEMIA | 56 | 91 | 147 |
| PROSTATE | 42 | 104 | 146 |
| BRAIN, CNS | 74 | 40 | 114 |
| BONE/SOFT TISSUE | 71 | 33 | 104 |
| KAPOSI'S SARCOMA | 89 | 12 | 101 |
| OTHER GI/PERITONEUM | 32 | 54 | 86 |
| HODGKIN'S LYPHOMA | 78 | 4 | 82 |
| OVARY | 26 | 47 | 73 |
| ANUS | 67 | 4 | 71 |
| ESOPHAGUS | 41 | 30 | 71 |
| MELANOMA | 25 | 40 | 65 |
| CORPUS UTERI | 18 | 46 | 64 |
| PANCREAS | 27 | 36 | 63 |
| KIDNEY | 25 | 36 | 61 |
| GENITAL SYSTEM, FEMALE | 48 | 7 | 55 |
| BLADDER | 20 | 33 | 53 |
| OTHERS | 41 | 8 | 49 |
| STOMACH | 16 | 29 | 45 |
| GENITAL SYSTEM, MALE | 42 | 1 | 43 |
| MYELOMA | 26 | 12 | 38 |
| SKIN, NON-MELANOMA | 23 | 15 | 38 |
| OTHER ENDOCRINE/NET | 30 | 7 | 37 |
| THYROID | 15 | 14 | 29 |
| OTHER URINARY SYSTEM | 18 | 5 | 23 |

*CNS* central nervous system, *GI* gastrointestinal, *NET* neuroendocrine tumor.

The increasing trend in the number of funded cancer research projects in Africa aligns with recent work by Virani and colleagues[18] in South Asia, consistent with rising levels of global funding and research for cancer[7,19]. Notwithstanding, global cancer research funding is still significantly tilted toward high-income countries (HICs)[6]. Within Africa, we observed notable disparities in the regional and sub-regional distribution of funded cancer research projects, with most projects concentrated in very few countries, particularly Egypt and South Africa. This is reflected in the large positive skew observed in our analysis, with the mean number of funded projects (63.35) much higher than the median value of 9 projects. Interestingly, the majority of studies from the CTG and WHO ICTRP databases were from Northern Africa. Because these two databases present investigator-reported clinical trials, the data likely reflect differences in domestic clinical and research infrastructure that exist between Northern Africa and sub-Saharan Africa (SSA). Conversely, most funded studies from SSA were from the ICRP and NIH World RePORT databases, populated by international partnerships, highlighting the dominance of externally driven cancer research in most SSA countries[9,10]. Sub-regionally, very few countries accounted for the vast majority of funded projects, and 16% of countries had no featured projects across all 4 databases. Our data closely mirror results

from a recent bibliometric analysis showing that Egypt and South Africa published nearly two-thirds of all cancer research output from Africa[9]. These disparities are attributable to uneven domestic funding and infrastructural deficits. Moreover, there is a tendency for HIC collaborators to partner with institutions/countries with already well-established research capacities and outputs, further perpetuating the inequity cycle[20,21].

Excluding Egypt, the overwhelming majority of funding sources across all databases in our analysis were from outside Africa. This is similar to findings by Davies et al. using the ICRP database for SSA[10], and Mutebi et al. through their bibliometric analysis of published papers from Africa[9]. The meager funding for research and development (R&D) by African governments impedes substantial progress in health. Globally, an estimated 90% of cancer research funding is by the government sector[7]. In 2007, African Union (AU) member states committed to devoting at least 1% of their annual gross domestic product (GDP) to R&D expenditure. However, based on available data to date, only Egypt (1.02% of GDP) has met this goal[22]. SSA countries average 0.44%, compared to the world average of 2.62%, and progress has been difficult to track for most African countries due to poor data availability[22,23].

This reliance on external funding is disturbing, especially because recent global occurrences have highlighted how volatile foreign aid can be, with sudden geopolitical shifts wiping away huge global health investments. For example, in early 2025, both the United Kingdom and the United States announced sudden cuts in global healthcare funding of up to £6 billion[24] and $10 billion[25], respectively. While African governments need to urgently address the funding gap, there is an equally important role for non-governmental actors within the continent. For example, charitable organizations such as the Bill and Melinda Gates Foundation and Cancer Research UK (CRUK) substantially contribute to research funding in the United States and the United Kingdom, respectively[6].

With over-reliance on external funding comes the risk of mismatched research priorities. Therefore, it is imperative to ensure the research agenda aligns with the locoregional cancer burden and needs. Consistent with a similar study on South Asia[18], most projects from both clinical trial databases examined in our study were interventional, likely reflecting the propensity of pharmaceutical companies and external funders to fund these kinds of studies. Notably, we found that funded projects focusing on cancer biology made up the least proportion of African projects in the ICRP database and there seems to be a slight but steady decrease in the relevance of biology-related research during the study period. This sharply contrasts with global data, where research on cancer biology received 41.2% of cancer funding investments between 2016 and 2020[6]. Cancer is a complex and heterogeneous disease, with distinct biological signatures in different populations[26–28]. Furthermore, African populations are the most genomically diverse in the world[29]. Therefore, it is important to invest in local research focusing on cancer biology. Conducting biological research often requires long-term commitment, training, and massive infrastructural investment, and this might explain the relative under-investment, especially since local research funding is scarce. Moreover, within the ICRP database, projects focusing on cancer research training were much fewer. Training and capacity building for local talent are essential for sustainable research progress and to stem the tide of "parachute" science. Dedicated training programs such as the Developing Excellence in Leadership, Training, and Science in Africa (DELTAS)[30] and the Initiative to Develop African Research Leaders (IDeAL)[31] programs have emerged in the African research scene but have mostly focused on infectious disease researchers. Similar programs focused on cancer research, such as the NCI's Global Training for Research and Equity in Cancer (GlobTREC) program[32], will help bridge this gap.

Breast cancer has the highest disease burden in Africa[1], and this correlates with receiving the most funding allocation from our analysis. Expectedly, cancers with infectious etiology also featured prominently across all databases analyzed, commensurate with their relatively high incidence rates in Africa[17,33]. Strikingly, our analysis suggests that several high-burden cancers, such as liver, prostate, and cervical cancers, remain underfunded despite their high mortality rates. A similar finding for

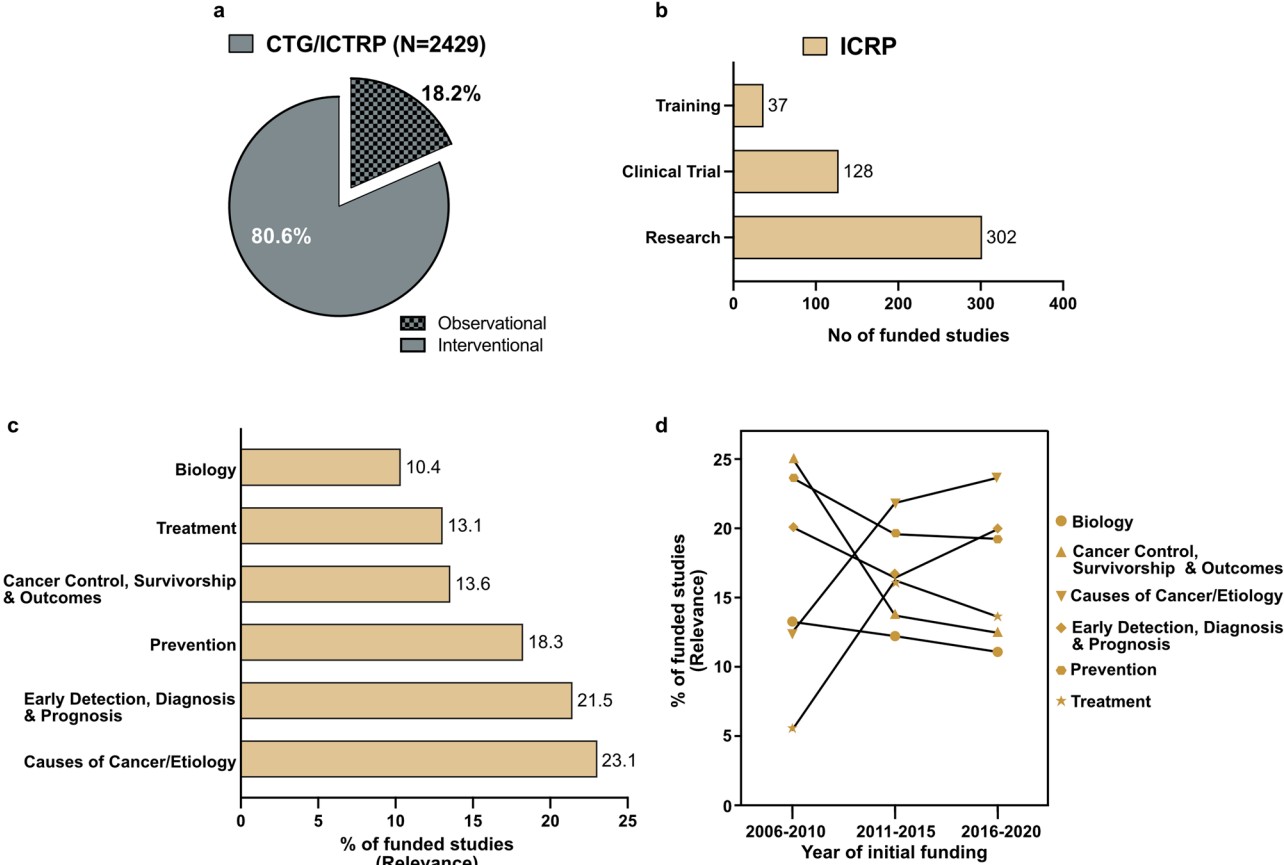

**Fig. 4 | Funded research projects by study types. a** CTG and WHO ICTRP reported studies classified as observational or interventional. **b** ICRP reported projects categorized into research, clinical trials, or training studies. **c** Percentage (relevance) of ICRP reported projects classified according to the Common Scientific Outline

(CSO). **d** A 5-yearly tracking of the percentage relevance of each ICRP CSO sub-category between 2006 and 2020. N.B. Individual studies in ICRP may fall under multiple study types/classifications.

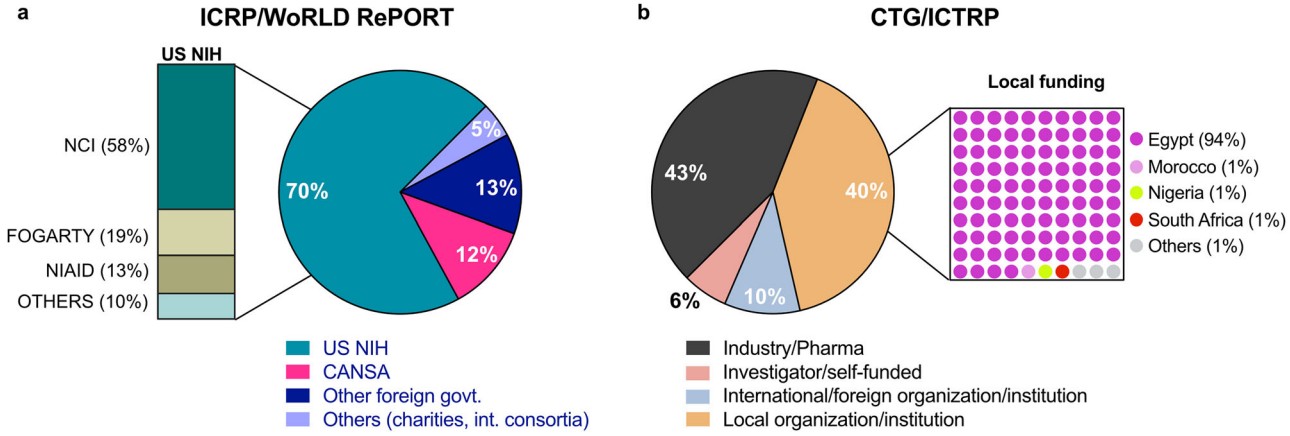

**Fig. 5 | Funding sources for cancer research projects in Africa. a** ICRP and NIH World RePORT funding sources. US NIH-funded studies zoomed out to show contributions from different agencies. **b** CTG and WHO ICTRP funding sources.

Locally funded studies zoomed out to show country-specific contributions. US NIH United States National Institutes of Health, NCI National Cancer Institute, NIAID National Institute of Allergy and Infectious Diseases.

prostate cancer was reported by Davies and colleagues[10]. This finding is concerning because men of African descent have an increased risk of prostate cancer, with an earlier onset of disease and an aggressive phenotype, resulting in poorer outcomes[34]. To improve cancer research funding in Africa, coordinated efforts towards research capacity building and training towards writing competitive grants are necessary. Further, intra-continental partnerships with local industries and philanthropic organizations must be

established and expanded. There is also an important role for National Cancer Control Plans (NCCPs) in developing appropriate mechanisms for the allocation of cancer research funding on the continent.

This study's key strength lies in its comprehensive design, integrating data from four major cancer research databases—ICRP, NIH World RePORT, CTG, and WHO ICTRP—offering detailed insights into research trends, funding sources, and regional disparities across Africa. By

**Fig. 6 | Correlation between funded cancer types and incidence/mortality rates.** Total number of funded studies per cancer type (green bars), plotted against incidence (blue dots) and mortality (red dots) data for each cancer type. Incidence and mortality data are derived from the International Agency for Research on Cancer (IARC) Global Cancer Observatory (GLOBOCAN) 2022 data.

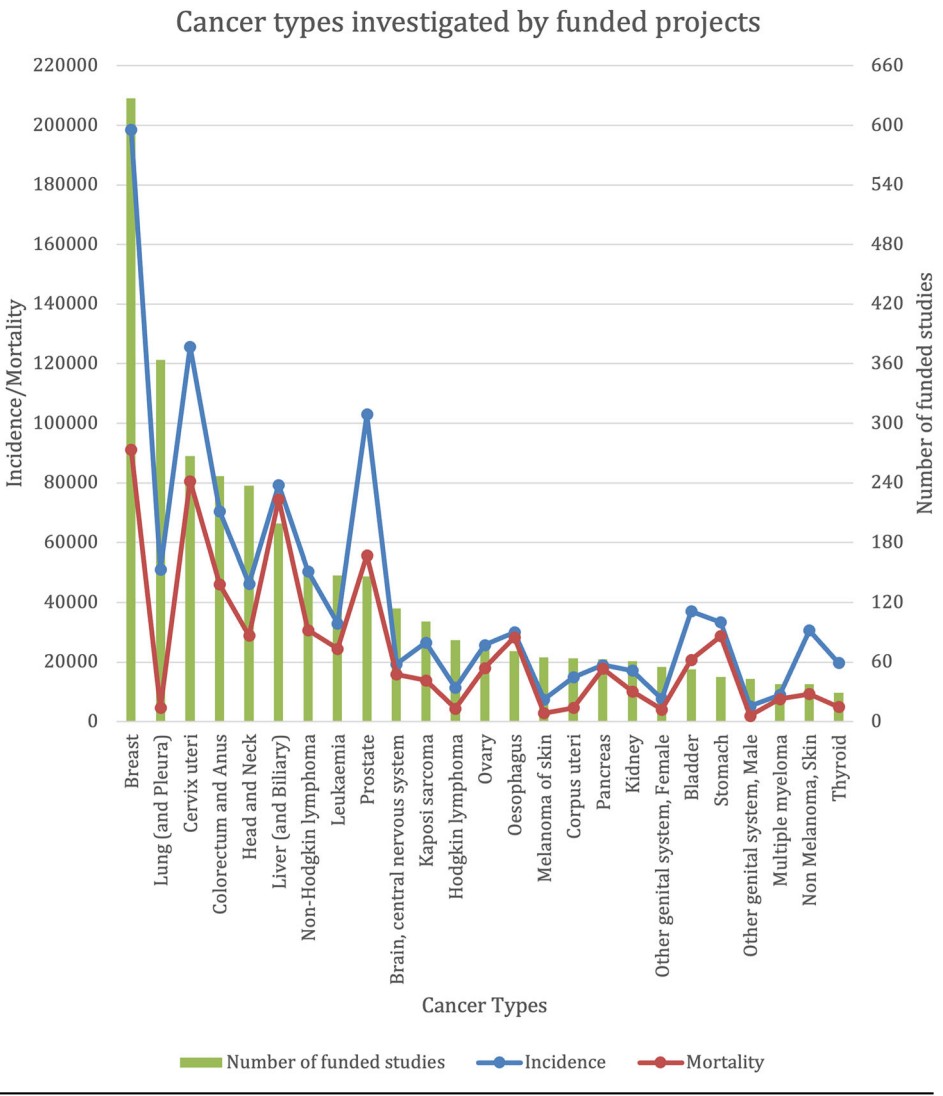

employing a multi-database approach, the study enhances data reliability and minimizes potential gaps, providing a robust foundation for understanding the cancer research funding landscape. The use of standardized classifications like the ICRP's Common Scientific Outline further strengthens the analysis by enabling consistent categorization of study types. Additionally, the inclusion of both clinical trials and observational studies expands the scope of the research, capturing diverse project types and funding streams. Despite these strengths, several limitations impact the study. Our findings reflect only the data reported in standard sources, excluding unreported funding, which limits the study's ability to capture the full scope of cancer research investments and may result in an incomplete understanding of the research landscape in Africa. Likewise, detailed information on the nature of affiliation of funded researchers with the African institutions is lacking, limiting our understanding of the dynamics of funding flow and administration. The reliance on reported data excludes unreported projects and non-cancer-related research funding, potentially overlooking other relevant contributions. The exclusion of professional development funding, such as conference travel, education, and travel grants, limits insights into capacity-building efforts. Potential overreporting remains a challenge due to overlapping records across databases, even with rigorous de-duplication efforts. Additionally, incomplete financial data across databases restricts in-depth analysis of funding amounts. Furthermore, the lack of visibility on intra-African collaborations may obscure key regional partnerships essential for local research capacity and resource-sharing initiatives. Finally, non-uniform

reporting structures across databases highlight the need for a centralized platform to ensure data consistency and transparency.

## Conclusions

This study offers the most comprehensive analysis to date of cancer research funding across Africa by synthesizing data from four major databases. While cancer research funding has steadily increased over the past two decades, major disparities persist. Funding remains concentrated in Northern Africa, particularly in Egypt and in South Africa, while many high-burden cancers, such as cervical, liver, and prostate cancers, are underfunded. Furthermore, the over-reliance on international funding sources highlights the need for greater domestic investment in cancer research, as contributions from African governments outside Egypt remain minimal. Limited focus on training and capacity-building programs further underscores the need to strengthen local research infrastructure to build sustainable expertise and reduce dependence on foreign collaborations. Future research should prioritize increased funding for underfunded high-burden cancers, particularly cervical and prostate cancers, to address unmet health needs. African governments should play a more proactive role in supporting cancer research to reduce dependency on external funding. Investment in capacity-building programs tailored to cancer research is essential for fostering local talent and infrastructure development. Promoting intra-African collaborations and partnerships with local industries and philanthropic organizations can enhance research output and diversify funding sources. To improve transparency and accountability, developing

centralized reporting platforms is crucial for tracking cancer research investments across the continent.

## Data availability
All data used in this study were derived from publicly available databases with descriptions and references provided in the paper. No primary data were generated for the study.

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

## Acknowledgements
We acknowledge all databases used in this study: ClinicalTrials.gov, World Health Organization International Clinical Trials Registry Platform, International Cancer Research Partnership Database, and National Institutes of Health World Research Portfolio Online Reporting Tools for generously making data available and attending to our questions during the conduct of this study. S.N.Z. receives partial salary support for research from the NIH/NCI Early-Stage Surgeon Scientist Program Grant P30 CA014520-48S4. The content of this publication does not necessarily reflect the views or policies of the Department of Health and Human Services, nor does mention of trade names, commercial products, or organizations imply endorsement by the US Government.

## Author contributions
S.N.F. and O.A. conceptualized and designed the study. E.M.G. contributed specific expertise in international cancer funding database management. O.A., O.E.D., and E.M.G. conducted data curation, deduplication, and data cleaning. O.A. and S.S.V. performed data analyses. O.A. and O.E.D. drafted the paper. S.N.F., E.M.G., A.R.S., A.A.O., and S.S.V. performed critical review and editing. All authors have read the paper draft and approved the submission.

## Competing interests
The authors declare no competing interests.
