## [Transparent Peer Review file · Communications Medicine]

Cancer Research Funding in Africa

Corresponding Author: Dr Syed Nabeel Zafar

Version 0:

Reviewer comments:

Reviewer #1

(Remarks to the Author)

The overall goal of this manuscript is laudable, with the potential to inform funders and researchers of needs for investment in cancer research infrastructure, cancer research and training in Africa. With that said, I have the following comments/questions:

- 1) Given that the authors are comparing funding vs. cancer burden, the authors should consider including a figure of cancer burden in Africa.
- 2) Related to cancer burden - authors should provide some additional information (2 - 3 sentences) on the quality of cancer data from Globocan. Globocan depends on the quality of cancer registries and percent coverage of the population. How well does Globocan cover burden of disease in those countries with little research funding?
- 3) The authors stated that they looked for investigators associated with an African institute in order to include the project. Are the authors able to determine whether the investigators has an adjunct position at the African Institute vs. primary position at the institute?
- 4) The authors need to do a better job explaining the numbers in their figures to limit the confusion for readers. For example, in Figure 1 - the author identified 3047 unique projects per the venn diagrams in A and B. In Figure 1C the total number of projects seems less (unless I am missing something here).
- 5) Related to this, Figure 2, authors should consider acknowledging that the sum of projects may be more due to one project being located in multiple locations in the figure description.
- 6) Since clinical trials is so different from cancer research, it would be advisable if the authors presented data from the two different types of databases separately instead of clumping them all together. This would be more informative.
- 7) For the types of cancer, authors should consider listing the cancers with the most number of projects (since this is the goal of the paper) rather than by alphabetical order.
- 8) Can the authors comment on the huge difference between the median and mean number of projects? I do not think this was discussed.

Reviewer #2

(Remarks to the Author)

The authors performed a retrospective review of funded cancer research projects conducted in Africa from January 2004 to December 2023 to generate a comprehensive overview of the current state of cancer research funding in Africa, highlight notable gaps and provide critical insights to guide data-driven decision-making. Utilizing four major databases of funded research projects, they identified 3047 unique projects. Most of these projects were in five countries, Egypt, South Africa, Nigeria, Kenya, and Uganda, and no projects were identified in 9 countries in Africa. Most funding was by outside institutions (NIH), and local funding was noted only in three countries (Egypt, Nigeria, and South Africa).

This report presents a useful summary of the research funding landscape that will be useful to both national and international policy makers, as well as to research scientists seeking funding. However, there are some issues that need to be considered to improve the manuscript.

Major

1. The authors cite several reports that have conducted similar reviews (PMID: 36356985, PMID: 37290022, PMID: 37269844, PMID: 33909474, etc etc). it would be helpful to revise the introduction to make a clearer case for yet another review – what does this review provide that cannot be discerned from the earlier papers?
2. The authors mention substantial variation in cancer burden, without giving one or two examples of the variation that is of utmost concern. They mention “staggering data and forecasts” without quantifying them; please include these details for clarity and emphasis.
3. The results of projects reported in the Venn diagrams confusingly. For each Venn diagram, the intersection represents the number of unique projects reported by both databases, while the number listed in each circle represents the number of projects reported by the respective database, but this number includes the projects reported in duplicate. Please check and correct.
4. Figure 1C – i wonder if it might be more informative to present the data by calendar year bars graph for each of the platforms (ICRP/ World Report and CTG/CTRP) with the bars side by side.
5. The paper presents funding information for viral- and non-viral cancers. Would it be possible for these to be defined clearly.
6. Figure 2A – would be possible to use gradient shading to show the number of projects funded in the respective countries in the region. The gradient can be based on the regional colors. I know this information is provided in Figure 3, but the spatial relationships of the countries is lost in the bar graphs.
7. Table 1 is sorted alphabetically – another way might be to sort according to frequency (? overall) or anatomic site (from head to feet, as is done in many cancer reports).
8. Figure 6 – it might be more striking to sort the bar graphs by number of funded studies from highest to lowest.
9. The report doesn't address issues of population counts across different countries/regions and demographic patterns or GDP. Although it may not be necessary to add this information to their analysis, maybe they should comment about why this aspect is not considered.

Minor comments

1. There is a typos in line 203, i.e., 444 projects should be 446, and another typo in line 204, i.e., 1717 projects should be 1716, based on data in the Venn diagrams.
2. There is a disconnect between the coding of study design – interventional versus observational as shown in Figure 4A and 4B. Interventional studies are clinical trials, yet this group of projects is not the majority as shown in Figure 4B. Moreover, Figure 4C also suggests that a smaller proportion of studies were dedicated to treatment. This needs to be clarified.

Version 1:

Reviewer comments:

Reviewer #1

(Remarks to the Author)

I would like to thank the authors for their responses and changes made to the manuscript. The revise manuscript presents a clearer picture. There are a few additional issues that should be addressed:

- 1) Please use the proper definition for "viral-associated cancers" for the sake of consistency within the field
- 2) Authors should include a note for the bar graph graph (Figure 1B), acknowledging that one of the databases (World RePORT) only has data from 2016.
- 3) In their response to reviewer #2, the authors stated that they included "studies on training". However, the overall goal of this manuscript is to look at cancer research funding, which is very different from funding training for cancer research. While related, cancer training is a different topic from cancer research. Is it appropriate to include projects supporting cancer training in this paper? Should this be included in the exclusion criteria?

Reviewer #2

(Remarks to the Author)

My concerns have been adequately addressed.

We are very grateful to the reviewers for their insightful comments and suggestions, which have helped improve the quality and clarity of our manuscript. Please find point-by-point responses to the comments below.

Reviewer #1 (Remarks to the Author):

The overall goal of this manuscript is laudable, with the potential to inform funders and researchers of needs for investment in cancer research infrastructure, cancer research and training in Africa. With that said, I have the following comments/questions:

1) Given that the authors are comparing funding vs. cancer burden, the authors should consider including a figure of cancer burden in Africa.

We thank the reviewer for this comment, which we duly considered. Given that we used cancer burden data directly from the IARC Global Cancer Observatory (GLOBOCAN) website and did not generate any secondary data from these, we have not included a dedicated figure on cancer burden. However, figure 6 includes cancer burden (incidence and mortality data derived from GLOBOCAN. We included the description and reference for the cancer burden data used in the Methods section (lines 180-184):

“As estimates of the burden of disease, we obtained data on age-standardized incidence and mortality rates as well as 5-year prevalence for specific cancer types in Africa, and for individual African countries, from the International Agency for Research on Cancer (IARC) Global Cancer Observatory (GLOBOCAN) 2022 data.¹⁶”

2) Related to cancer burden - authors should provide some additional information (2 - 3 sentences) on the quality of cancer data from Globocan. Globocan depends on the quality of cancer registries and percent coverage of the population. How well does Globocan cover burden of disease in those countries with little research funding?

Thank you for the comment and suggestion. We have now included this in the introduction (lines 104-108):

“Resource allocation and funding decisions rely on the global cancer burden data (GLOBOCAN), which depend on well-established cancer registries. Consequently, the lack of established cancer registries in several African countries, as well as the deficiencies in quality and coverage of the existing registries, means that the data estimates might not adequately reflect the cancer burden on the continent.⁸”

3) The authors stated that they looked for investigators associated with an African institute in order to include the project. Are the authors able to determine whether the investigators has an adjunct position at the African Institute vs. primary position at the institute?

Our search was primarily based on the project including at least one African country as one of the study locations/sites. Where the database permits specification, we included all studies, whether the primary or the collaborating investigator(s) is affiliated with an African institution, as long as they satisfied the primary requirement. However, we are unable to determine the nature of affiliation of the investigators from the databases. We acknowledge that this is a limitation of the study and included this in the “Strengths and limitations”

section of the revised manuscript. We have rewritten the Methods to include a well-defined section on “Inclusion and Exclusion criteria” and reworded the Search Strategy for better clarity.

4) The authors need to do a better job explaining the numbers in their figures to limited the confusion from readers. For example, in Figure 1 - the author identified 3047 unique projects per the venn diagrams in A and B. In Figure 1C the total number of projects seems less (unless I am missing something here).

We have rewritten the Results to better explain the Venn diagram (lines 230-236):

“We identified 402 funded projects from the ICRP, 446 from WoRLD RePORT, 1,653 from CTG, and 1,716 from the ICTRP database over the 20 years (Fig. 1A). Following deduplication by unique NIH 8-digit IDs, we identified 230 projects overlapping between both ICRP and WoRLD RePORT, resulting in 618 unique projects between these two databases (Fig. 1A, left). Similarly, querying by unique NCT IDs revealed 940 overlapping projects between CTG and ICTRP, resulting in 2,429 unique projects from these databases (Fig. 1A, right). Altogether, our search generated 3,047 unique funded projects across all 4 databases (Fig. 1A).”

We have also adjusted the figure to better annotate what each number indicates (see new Figure 1 below and in the updated manuscript). Regarding Figure 1C (new Figure 1B), we aimed to show data for 3 consecutive 5-year intervals from 2006 to 2020 to demonstrate the increasing trend, therefore, the total number in 1C is less than the number from the full duration covered by the study (2004-2023).

(New) Figure 1. Number of funded cancer projects in African countries from (A, left) ICRP and World RePORT, (A, right) CTG and WHO ICTRP queried

through the 20-year study period (B) 5-year trends in the number of cancer projects from all four databases between 2006 and 2020. (ICRP, International Cancer Research Partnership; WoRLD RePORT, National Institutes of Health World Research Portfolio Online Reporting Tools; CTG, ClinicalTrials.gov; ICTRP, World Health Organization International Clinical Trials Registry Platform).

5) Related to this, Figure 2, authors should consider acknowledging that the sum of projects may be more due to one project being located in multiple locations in the figure description.

*This has now been included in the Figure legend (see lines 278-280):
 “N.B. The sum of the numbers shown in A here (3,611) is more than the total number of studies in Figure 1A (3,047), as some studies involve multiple countries cutting across different subregions.”*

6) Since clinical trials is so different from cancer research, it would be advisable if the authors presented data from the two different types of databases separately instead of clumping them all together. This would be more informative.

We thank the reviewer for this comment, which we agree with. Throughout the manuscript, we’ve separated the data from the two different types of databases where possible- see Figure 1 (A&B), Figure 2B, Table 1, Figure 4, and Figure 5.

7) For the types of cancer, authors should consider listing the cancers with the most number of projects (since this is the goal of the paper) rather than by alphabetical order.

This has now been done- please see the new Table 1.

	ICRP	CTG/ICTRP	TOTAL
BREAST	80	547	627
LIVER (AND PLEURAL)	104	260	364
NOT SITE-SPECIFIC CANCER	168	187	355
CERVIX UTERI	143	124	267
HEAD & NECK	124	113	237
LIVER (AND BILLIARY)	88	111	199
COLON & RECTAL	38	138	176
NON-HODGKIN'S LYMPHOMA	96	58	154
LEUKAEMIA	56	91	147
PROSTATE	42	104	146
BRAIN, CNS	74	40	114
BONE/SOFT TISSUE	71	33	104
KAPOSI'S SARCOMA	89	12	101
OTHER GI/PERITONEUM	32	54	86
HODGKIN'S LYPHOMA	78	4	82
OVARY	26	47	73
ANUS	67	4	71
OESOPHAGUS	41	30	71
MELANOMA	25	40	65
CORPUS UTERI	18	46	64

PANCREAS	27	36	63
KIDNEY	25	36	61
GENITAL SYSTEM, FEMALE	48	7	55
BLADDER	20	33	53
OTHERS	41	8	49
STOMACH	16	29	45
GENITAL SYSTEM, MALE	42	1	43
MYELOMA	26	12	38
SKIN, NON MELANOMA	23	15	38
OTHER ENDOCRINE/NET	30	7	37
THYROID	15	14	29
OTHER URINARY SYSTEM	18	5	23

Table 1. Cancer types investigated by funded projects (A) aggregated across all 4 databases and (B) within each database. *CNS*, Central Nervous System; *GI*, Gastrointestinal; *NET*, Neuroendocrine tumor.

8) Can the authors comment on the huge difference between the median and mean number of projects? I do not think this was discussed.

This has now been included in the discussion (lines 388-390): “However, there are stark regional disparities, with most projects concentrated in Northern Africa, particularly Egypt and South Africa, which together account for almost two-thirds of Africa’s research output. This is reflected in the large positive skew observed in our analysis, with the mean number of funded projects (63.35) much higher than the median value of 9 projects.”

Reviewer #2 (Remarks to the Author):

The authors performed a retrospective review of funded cancer research projects conducted in Africa from January 2004 to December 2023 to generate a comprehensive overview of the current state of cancer research funding in Africa, highlight notable gaps and provide critical insights to guide data-driven decision-making. Utilizing four major databases of funded research projects, they identified 3047 unique projects. Most of these projects were in five countries, Egypt, South Africa, Nigeria Kenya, and Uganda, and no projects were identified in 9 countries in Africa. Most funding was by outside institutions (NIH), and local funding was noted only in three countries (Egypt, Nigeria, and South Africa).

This report presents a useful summary of the research funding landscape that will be useful to both national and international policy makers, as well as to research scientists seeking funding. However, there are some issues that need to be considered to improve the manuscript.

Major

1. The authors cite several reports that have conducted similar reviews (PMID: 36356985, PMID: 37290022, PMID: 37269844, PMID: 33909474, etc etc). it would be helpful to revise the introduction to make a clearer case for yet another review – what does this review provide that cannot be discerned from the earlier papers?

We thank the reviewer for this comment. We have revised the introduction to better highlight this (see lines 110-115): “Previous works have provided useful

insights into the state of cancer research funding in Africa; however, these have either been through content analysis of funding acknowledgments in published studies or have focused on a singular database.^{6,7,9,10} Therefore, here we reviewed data from four public databases with both unique and overlapping characteristics to achieve a more comprehensive view of the landscape of cancer research funding on the continent.”

2. The authors mention substantial variation in cancer burden, without giving one or two examples of the variation that is of utmost concern. They mention “staggering data and forecasts” without quantifying them; please include these details for clarity and emphasis.

*We thank the reviewer for this comment. An example of the variation in cancer burden has now been included in the introduction (lines 83-87):
“The disease burden and mortality vary considerably across organ systems, geographical regions, environments, and resource settings. For example, although the incidence rates for breast cancer in North American and European countries can be up to four times higher than in Western African countries, West Africa accounts for some of the highest mortality rates seen with the disease.^{1”}*

Moreover, we have attempted to improve the entire introduction for clarity.

3. The results of projects reported in the Venn diagrams confusingly. For each Venn diagram, the intersection represents the number of unique projects reported by both databases, while the number listed in each circle represents the number of projects reported by the respective database, but this number includes the projects reported in duplicate. Please check and correct.

We have rewritten the Results to better explain the Venn diagram (lines 230-236):

“We identified 402 funded projects from the ICRP, 446 from WoRLD RePORT, 1,653 from CTG, and 1,716 from the ICTRP database over the 20 years (Fig. 1A). Following deduplication by unique NIH 8-digit IDs, we identified 230 projects overlapping between both ICRP and WoRLD RePORT, resulting in 618 unique projects between these two databases (Fig. 1A, left). Similarly, querying by unique NCT IDs revealed 940 overlapping projects between CTG and ICTRP, resulting in 2,429 unique projects from these databases (Fig. 1A, right). Altogether, our search generated 3,047 unique funded projects across all 4 databases (Fig. 1A).”

We have also adjusted the figure to better annotate what each number indicates (see new Figure 1 below and in the updated manuscript).

(New) Figure 1. Number of funded cancer projects in African countries from (A, left) ICRP and World RePORT, (A, right) CTG and WHO ICTRP queried through the 20-year study period (B) 5-year trends in the number of cancer projects from all four databases between 2006 and 2020. (ICRP, International Cancer Research Partnership; WoRLD RePORT, National Institutes of Health World Research Portfolio Online Reporting Tools; CTG, ClinicalTrials.gov; ICTRP, World Health Organization International Clinical Trials Registry Platform).

4. Figure 1C – I wonder if it might be more informative to present the data by calendar year bars graph for each of the platforms (ICRP/ World Report and CTG/CTRP), with the bars side by side.

We thank the reviewer for this comment, which we duly considered. Given that we have four databases reviewed over 20 years we thought that representing calendar years for each database might be cumbersome and difficult to assimilate for readers. Also, because our deduplication steps used unique study IDs, we treated studies from the two different types of databases (i.e., Type I- ICRP/World RePORT and Type II- CTG/ICTRP) as a unit for downstream analyses. This allowed us to avoid double-counting the duplicates for our subsequent analyses. Moreover, one of the databases (World RePORT) only has data from 2016, therefore, we thought the best way to showcase the increasing trend would be through the consolidated 5-year bar graphs.

5. The paper presents funding information for viral- and non-viral cancers. Would it be possible for these to be defined clearly.

Thank you for the comment- this has now been clearly defined (lines 209-211):

“Where mentioned, the term “Viral-associated cancers” was used to refer to cancers of the Cervix, Head and Neck, Liver, Anus, Non-Hodgkin’s Lymphoma, and Kaposi’s Sarcoma.”

6. Table 1 is sorted alphabetically – another way might be to sort according to frequency (? overall) or anatomic site (from head to feet, as is done in many cancer reports).

*Thank you for this comment. This has now been done- please see **new Table 1**.*

7. Figure 6 – it might be more striking to sort the bar graphs by number of funded studies from highest to lowest.

We thank the reviewer for this comment. This has now been revised- see new Figure 6 below and in the updated manuscript.

Figure 6. Correlation between number of funded studies by cancer types and incidence and mortality rates. CNS, central nervous system.

Minor comments

1. There is a typos in line 203, i.e., 444 projects should be 446, and another typo in line 204, i.e., 1717 projects should be 1716, based on data in the Venn diagrams.

Thank you for the comments- all have now been corrected (see lines 230 to 236):

“We identified 402 funded projects from the ICRP, 446 from WoRLD RePORT, 1,653 from CTG, and 1,716 from the ICTRP database over the 20 years (Fig. 1A). Following deduplication by unique NIH 8-digit IDs, we identified 230 projects overlapping between both ICRP and WoRLD RePORT, resulting in 618 unique projects between these two databases (Fig. 1A, left).

Similarly, querying by unique NCT IDs revealed 940 overlapping projects between CTG and ICTRP, resulting in 2,429 unique projects from these databases (Fig. 1A, right). Altogether, our search generated 3,047 unique funded projects across all 4 databases (Fig. 1A)."

2. There is a disconnect between the coding of study design – interventional versus observational, as shown in Figures 4A and 4B. Interventional studies are clinical trials, yet this group of projects is not the majority as shown in Figure 4B. Moreover, Figure 4C also suggests that a smaller proportion of studies were dedicated to treatment. This needs to be clarified.

We thank the reviewer for this comment.

Figure 4A refers to only CTG and ICTRP data- these databases classify projects as either Interventional or Observational (see lines 285-290): "The CTG and WHO ICTRP databases categorize projects into either interventional or observational study types. Across both databases, the majority of the reported projects (80%) were interventional (Fig. 4A). CTG provides additional granular data on the study designs for reported projects. From these, we found that 79% of the interventional studies reported were randomized clinical trials, and all reported observational projects were prospective cohort studies."

Figures 4B and 4C, on the other hand, refer only to ICRP data, which uses a different classification system as "Clinical Trial," "Research," or "Training," and also provides additional coding information into the CSO scientific domains (see lines 290-297):

"The ICRP database classifies reported studies into either "Clinical Trial", "Research" or "Training" project types, with some studies including a combination of these. Most of the reported studies from Africa fell into the Research category (302), followed by Clinical Trial (128), with fewer studies on Training (37) (Fig. 4B). The ICRP CSO allows further classification of funded projects into six relevant scientific domains. Over the 20-year period in ICRP, 23.1% of funded projects in Africa were on causes of cancer/etiology, 21.5% on early detection, diagnosis, and prognosis, 18.3% on cancer prevention, 13.6% on cancer control, survivorship, and outcomes, 13.1% on treatment, and 10.4% on biology (Fig. 4C, Left).

We are very grateful to the reviewers for their final review and additional suggestions. Please find point-by-point responses to the comments below.

Reviewer #1 (Remarks to the Author):

I would like to thank the authors for their responses and changes made to the manuscript. The revised manuscript presents a clearer picture. There are a few additional issues that should be addressed:

1) Please use the proper definition for "viral-associated cancers" for the sake of consistency within the field

We thank the review for this comment, we have further clarified the definition of "viral-associated cancers" backed by literature

Where mentioned, the term "Viral-associated cancers" was used to refer to cancers with etiology prominently linked to viral agents as defined in the literature.¹⁷ These include cancers of the Cervix, Head and Neck, Liver, Anus, and Stomach, as well as Non-Hodgkin's Lymphoma and Kaposi's Sarcoma.

Also, see additional reference added to support the definition (no 17 in reference list):

Plummer M, de Martel C, Vignat J, Ferlay J, Bray F, Franceschi S. Global burden of cancers attributable to infections in 2012: a synthetic analysis. The Lancet Global Health. 2016;4(9):e609-e616. doi:10.1016/S2214-109X(16)30143-7.

2) Authors should include a note for the bar graph (Figure 1B), acknowledging that one of the databases (World RePORT) only has data from 2016.

This has been added. Thank you. See new Figure 1 legend

Figure 1. Total number of studies and 5-year trends of funded cancer projects in African countries across all four databases. (a) Total search results from each database through the 20-year study period are shown. Deduplication of overlapping projects between ICRP/World RePORT and between CTG/WHO ICTRP resulted in 3047 projects across all databases taken forward for subsequent analyses. (b) 5-year trends in the number of cancer projects from all four databases between 2006 and 2020. *ICRP, International Cancer Research Partnership; WoRLD RePORT, National Institutes of Health World Research Portfolio Online Reporting Tools; CTG, ClinicalTrials.gov; ICTRP, World Health Organization International Clinical Trials Registry Platform. N.B. The NIH World RePORT database only includes data from 2016 onwards.*

3) In their response to reviewer #2, the authors stated that they included "studies on training". However, the overall goal of this manuscript is to look at cancer research funding, which is very different from funding training for cancer research. While related, cancer training is a different topic from cancer research. Is it appropriate to include projects supporting cancer training in this paper? Should this be included in the exclusion criteria?

We appreciate the reviewer's comment on this. The International Cancer Research Partnership (ICRP) database included data on funded projects for cancer research training. We decided to include these studies because, in the context of global oncology funding, most training grants include actual research projects as well, often funding an early career researcher to conduct a project while receiving longitudinal training alongside, or funding established researchers/institutions to mentor and support early career researchers on individual projects. Moreover, in the case that these awards don't directly support a research project, they would ultimately be expected to lead to increased cancer research capacity and output, which fits into the overall goal of our manuscript.

Reviewer #2 (Remarks to the Author):

My concerns have been adequately addressed.